# High Incidence of Obstetric Anal Sphincter Injuries among Immigrant Women of Asian Ethnicity

**DOI:** 10.3390/jcm12031044

**Published:** 2023-01-29

**Authors:** Yoav Baruch, Ronen Gold, Hagit Eisenberg, Hadar Amir, Lee Reicher, Yariv Yogev, Asnat Groutz

**Affiliations:** 1Urogynecology and Pelvic Floor Unit, Department of Obstetrics and Gynecology, Tel Aviv Medical Center, Faculty of Medicine, Tel Aviv University, Tel Aviv 6997801, Israel; 2Faculty of Medicine, Tel Aviv University, Tel Aviv 6997801, Israel; 3Lis Maternity Hospital, Tel Aviv Medical Center, Faculty of Medicine, Tel Aviv University, Tel Aviv 6997801, Israel

**Keywords:** OASI, obstetric anal sphincter injury, Asian ethnicity, Caucasian ethnicity

## Abstract

(1) Background: Obstetric anal sphincter injuries (OASI) may complicate vaginal deliveries. The aim of the present study was to explore the incidence and clinical characteristics of OASI among Asian women living in a Western country compared to local Caucasian women. (2) Methods: A retrospective cohort study of 380 women diagnosed with OASI, following singleton vaginal deliveries, during a 10-year period (January 2011 to December 2020). Exclusion criteria: age < 18 years, stillbirth, and breech presentation. Demographic, clinical, and obstetrical data were obtained, and a comparison between Asian and Caucasian women was performed. (3) Results: There were 35 cases of OASI among 997 women of Asian ethnicity compared to 345 cases of OASI among 86,250 Caucasian women (3.5% vs. 0.4%, respectively, *p* < 0.001). Asian women endured a significantly higher rate of fourth-degree OASI (17.1%) even though they bore smaller newborns (3318 g vs. 3501 g, *p* = 0.004), and birth weights rarely exceeded 3800 g (2.8% vs. 25.8%, *p* < 0.001). Asian ethnicity was also associated with a significantly higher risk for blood transfusion following OASI and a lower tendency for postpartum follow up. (4) Conclusions: Immigrant women of Asian ethnicity had a nine-fold higher rate of OASI, much higher than previously reported. Furthermore, Asian women had higher rates of fourth-degree OASI.

## 1. Introduction

Obstetric anal sphincter injuries (OASI) may complicate vaginal deliveries and may be associated with substantial long-term functional sequelae, such as perineal pain or dyspareunia and/or anal incontinence. In addition, OASI may also have a negative impact on social and psychological well-being; harm mother’s relationships with her newborn, partner, or other family members; and result in long-term psychological illnesses such as anxiety and depression [1,2]. The incidence of OASI varies considerably, from 0.5% to 6%, for reasons that are not fully understood [3,4].

Risk factors for OASI include primiparity, operative vaginal delivery, fetal occipito-posterior position, high birthweight, prolonged second stage of labor, short perineal body, and midline episiotomy [5,6,7]. Racial variations, particularly Asian maternal ethnicity, have also been linked to OASI. Researchers have found that Asian women are more likely to experience OASI, with an increased risk of 1.5–4.6 fold [8,9,10,11,12]. Additionally, Asian ethnicity was reported in a Canadian study to be an independent risk factor for OASI [13]. A systematic review conducted by Wheeler et al. suggested that the increased OASI risk is limited to Asian women living in Western countries and has not been noted for Asian women living in Asia [14]. It is assumed that migration-specific factors account for at least some of the observed trends.

The aim of the present study was to explore the incidence and clinical characteristics of OASI among Asian women living in a Western country compared to the local Caucasian women.

## 2. Materials and Methods

This is a retrospective cohort study of 380 women diagnosed with OASI, following singleton vaginal deliveries, during a 10-year period (January 2011 to December 2020). The study was conducted in a tertiary university-affiliated hospital (Lis Maternity Hospital, Tel Aviv Medical Center, Tel Aviv University, Israel) with more than 12,000 deliveries per year. The study protocol was reviewed and approved by the institutional ethical review board. Asian ethnicity was defined as having been born in one of the following countries: Cambodia, China, Mongolia, Nepal, Philippines, Japan, Singapore, Sri Lanka, South Korea, Thailand, or Vietnam.

Demographic, maternal, obstetric, and neonatal parameters were compared between OASI cases diagnosed among local Caucasian women versus immigrant women of Asian ethnicity. All data were retrieved from a computerized database using the International Classification of Diseases 9th Revision codes for perineal lacerations. Exclusion criteria composed of age < 18 years, stillbirths, breech presentation, or ethnicity other than Caucasian or Asian. Comorbidities, use of drugs, alcohol, smoking, gestational diabetes status, and weight gain during pregnancy were among the maternal characteristics examined. Body mass index (BMI) was calculated from height and weight measurements prior to pregnancy or during the first trimester. Obstetric-related details included parity, gestational age, intrapartum fever, length of the first and second stages of labor, usage of oxytocin for the induction or augmentation of labor, fetal presentation and position, episiotomy, instrumental-assisted vaginal delivery, and incidence of shoulder dystociaNeonatal characteristics included gender, Apgar scores, and birth weights. Maternal complications included: OASI, blood loss, and the need for blood-product transfusions. 

At our institution, midwives handle uncomplicated deliveries. Instrumental-assisted deliveries are performed only by obstetricians using a vacuum device. Vacuum extraction is performed when the fetal head is stationed below the level of the ischial spines (at +1 or more). For nulliparous women, a prolonged second stage is defined as two hours or longer, and for parous women, one hour or longer, with an additional one hour if epidural analgesia is employed. A selective episiotomy policy is employed, and only mediolateral episiotomies are performed. Perineal protection is always undertaken during delivery, either spontaneous or instrumental. Diagnosis of the severity of obstetric tears is performed by obstetricians. According to the American College of Obstetrics and Gynecology (ACOG) guidelines, a third-degree perineal tear involves the anal sphincter muscle, while a fourth-degree perineal tear extends through the anal sphincter and into the rectal mucosa [15]. Repair of OASI is performed only by senior Obstetricians who have completed a training course for the diagnosis and repair of OASI. Repair of a fourth-degree OASI is performed by colorectal specialists.

Women who endure OASI are evaluated at the urogynecological clinic at 1, 6, and 12 months after delivery. Pelvic floor muscle training is strongly recommended during the puerperium period. Transrectal ultrasound (TRUS) and manometry are performed during the first year after delivery in order to assess the integrity and function of the anal sphincter.

## 3. Statistical Analysis

Categorical variables are presented as frequency and percentage. Continuous variables were evaluated for normal distribution using a histogram and quantile–quantile plots. Variables that were normally distributed are summarized as mean and standard deviation, while skewed variables are presented as median and interquartile range. Categorical variables were compared between the two groups using the chi-square test or Fisher’s exact test. Continuous variables and ordinal variables were compared using an independent samples *t*-test or Mann–Whitney test. A multivariable logistic regression analysis was performed to evaluate the association between fourth-degree OASI and Asian ethnicity while controlling for potential confounders. All statistical tests were two-sided, and a *p*-value less than 0.05 was considered statistically significant. SPSS software (IBM SPSS statistics, version 27, IBM Corporation, Armonk, New York, NY, USA, 2020) was used for all statistical analyses.

## 4. Results

The study population included 997 women of Asian ethnicity and 86,250 Caucasian women who met the inclusion criteria. There were 35 cases of OASI among women of Asian ethnicity compared to 345 cases among Caucasian women, reflecting a nine-fold higher rate of OASI among Asian women (3.5% vs. 0.4%, respectively, *p* = 0.001). Among the local Caucasian women, 42% were primiparous, 0.7% of whom experienced OASI. Among women of Asian ethnicity, 48% were primiparous, 4.8% of whom experienced OASI.

The maternal and neonatal characteristics of Caucasian versus Asian women who endured OASI are presented in Table 1. Most obstetric parameters were similar in both groups except for a lower rate of epidural analgesia among women of Asian ethnicity (48.6% vs. 76%, *p* < 0.001). Women of Asian ethnicity endured more extensive perineal injury with a significantly higher proportion of fourth-degree OASI compared to Caucasian women (17.1% vs. 6.6%, *p* = 0.39), although they bore smaller newborns (mean birth weight of 3318 g vs. 3501 g, *p* = 0.004) and barely delivered babies of birth weights exceeding 3800 g (2.8% vs. 25.8%, *p* < 0.001). Additionally, women of Asian ethnicity were less likely to complete the recommended postpartum follow up (17.1% vs. 38%, *p* = 0.012).

Differences in maternal blood loss are presented in Table 2. Asian ethnicity was associated with significantly higher blood loss, represented by the change in hemoglobin levels from baseline (4.2 g/dL vs. 2.6 g/dL, *p* = 0.005).

A multivariable logistic regression analysis of risk factors for fourth-degree OASI is presented in Table 3. Of all factors, only Asian ethnicity was found to be significantly associated with fourth-degree OASI (OR 2.9; 95% C.I. 1.02–8.35).

## 5. Discussion

Results of the present study show that, in comparison to local Caucasian women, immigrant women of Asian ethnicity had up to nine-fold incidence of OASI, despite having smaller neonates. Furthermore, Asian women experienced more extensive tears with an increased prevalence of fourth-degree OASI. This may also explain why Asian women lost more blood and had significantly lower postpartum hemoglobin levels. The vast majority of known risk factors investigated in the study were similar in the two study groups. However, a multivariable logistic regression analysis found that of various known risk factors for OASI, such as primiparity, epidural analgesia, birth weight, vacuum instrumental delivery, episiotomy, and occipito-posterior position of the fetus, only Asian ethnicity was associated with fourth-degree OASI.

The reasons for the high incidence of OASI among women of Asian ethnicity living in Western countries are not fully understood. Some anatomical mechanisms, such as shortened perineal body length, were suggested to explain the higher rate of OASI among Asian women [2]. However, measurements of perineal body length prior to delivery were not found to be a predictor parameter for OASI [16,17,18]. Numerous studies have linked high birth weights to OASI. Maternal lifestyle may influence in utero growth, as Asian women living in Western counties were found to have higher rates of gestational diabetes mellitus and macrosomia [19]. Additionally, variances in birth weights among Asian women residing in non-Asian countries can also be attributed to differences in male partner ethnicity [20]. Results of the present study show that Asian women, in comparison with Caucasian women, bore smaller newborns (3318 g vs. 3501 g, *p* = 0.004) and barely delivered babies of birth weights exceeding 3800 g (2.8% vs. 25.8%, *p* < 0.001). We, therefore, used a cut-off of 3800 g as a threshold for fetal overgrowth among Asian women. This observation reinforces the importance of using population-specific fetal weight nomograms and that the diagnosis of fetal macrosomia should not rely on an absolute value that is clearly not suitable for all populations.

The role of epidural analgesia as an independent risk factor for OASI was investigated in several studies with conflicting results [21,22,23]. In addition, some studies have shown that immigrant women are less likely to choose epidural analgesia [24,25,26]. In a survey conducted in four obstetrical clinics in Germany, fear and language barriers were the two main factors contributing to a decreased use of epidural analgesia among immigrant women [27]. In the present study, the local Caucasian population was more inclined to choose epidural analgesia compared to Asian women, even though epidural analgesia was offered free of charge to all women. However, a statistical analysis failed to prove or rule out the significance of epidural analgesia in the pathophysiology of OASI.

Communication during birth between the obstetrician or midwife and the parturient is essential, and “a lack of communication” was also found to be a risk factor for OASI [28]. The importance of communication is reinforced by the fact that actively pushing as the fetal head is crowning may increase the risk of OASI [29]. A language barrier between caregivers and immigrants is one of the main reasons for the poor quality of communication [30]. In a Swedish study, Esscher et al. demonstrated that foreign-born women were at a higher risk for maternal and neonatal morbidity and mortality during delivery [31]. Furthermore, a language barrier was found to be an independent risk factor of OASI [32]. In the present study, exact data regarding language barriers were not available; however, communication with most Asian women was in English, which is usually not the native language of most patients and midwives. The lower tendency to complete follow up after the puerperium period among women of Asian ethnicity is another significant observation in the present study. This may be at least in part attributed to the language barrier. Offering leaflets in the native language could be a potential intervention to increase compliance rates.

The study’s limitations include its retrospective nature and the lack of data regarding the male partners and the socioeconomic status of the women. The strengths of the study are the long period of time covered, the relatively high number of patients, and that all deliveries were from a tertiary university-affiliated hospital with more than 12,000 deliveries per year. The nine-fold higher rate of OASI among women of Asian ethnicity is among the highest reported to date, and if adjusted for fetal weight, the magnitude of this high rate may even rise. The reasons for these findings remain elusive and merit further investigation using large cohorts.

## 6. Conclusions

In the present cohort, the incidence of OASI was up to nine times higher among Asian women living in a Western country, much higher than previously reported. Furthermore, women of Asian ethnicity had higher rates of fourth-degree tears and a lower tendency to complete the recommended postpartum follow up.

## Figures and Tables

**Table 1 jcm-12-01044-t001:** Maternal and neonatal characteristics of Caucasian women versus women of Asian ethnicity who endured OASI.

Mean ± Standard Deviation; or Median (Interquartile Range); or *N* (%)	Caucasian Women with OASI (*N* = 345)	Asian Women with OASI (*N* = 35)	*p*
**OASI parameters**			
Rate of OASI	0.40%	3.51%	<0.001
Third-degree tear	322 (93.33%)	29 (82.9%)	0.039
Fourth-degree tear	23 (6.66%)	6 (17.1%)	
**Maternal characteristics**			
Age (years)	31.7 (±4.5)	33 (±3.7)	NS
Primiparity	254 (73.6%)	23 (65.7%)	NS
Diabetes (gestational or pregestational)	18 (5.2%)	2 (5.7%)	NS
BMI at first pregnancy visit (kg/m^2^)	21.6 (19.5–24.3)	22.2 (20.5–24.6)	NS
**Intrapartum**			
Gestational week at delivery	40 (39.2–40.8)	39.8 (38.6–40.4)	NS
Epidural analgesia	265 (76.8%)	17 (48.6%)	0.001
Medio-lateral episiotomy	114 (33%)	17 (46%)	NS
Occipito-posterior position	26 (7.5%)	1 (2.9%)	NS
Use of Pitocin (induction/augmentation)	222 (64.3%)	17 (48.6%)	NS
Prolonged second stage	101 (29.2%)	7 (20%)	NS
Vacuum-assisted delivery	85 (24.6%)	10 (28.6%)	NS
**Neonatal**			
Birth weight (grams)	3501 (±419)	3318 (±329)	0.004
Macrosomia (≥4000 g)	43 (13%)	1 (2.8%)	NS
Birth weight ≥ 3800 g	89 (25.8%)	1 (2.8%)	<0.001
Male gender	213 (61.7%)	19 (54.3%)	NS

NS = statistically nonsignificant, *p* > 0.05.

**Table 2 jcm-12-01044-t002:** Differences in OASI-related blood loss complications among Caucasian women and women of Asian ethnicity.

Median (Interquartile Range); or *N* (%)	Caucasian Women with OASI (*N* = 345)	Asian Women with OASI (*N* = 35)	*p*
**Maternal blood loss parameters**			
HB before delivery (g/dL)	12.5 (11.6–13)	12.9 (12–13.6)	0.028
Decrease in HB from baseline (g/dL)	2.6 (0.9–4.2)	4.2 (2.4–5.5)	0.005
HB level ≤ 7 (g/dL)	32 (9.3%)	5 (14.3%)	NS
Need for packed-cell transfusion	44 (12.75%)	6 (17.1%)	NS

HB = Hemoglobin. NS = statistically nonsignificant, *p* > 0.05.

**Table 3 jcm-12-01044-t003:** A multivariable logistic regression analysis of risk factors for fourth-degree OASI.

**Parameters**	OR (95% C.I.)	*p*-Value
Asian ethnicity	2.9 (1.02–8.35)	0.047
Epidural analgesia	0.83 (0.33–2.08)	0.694
Birth weight	1.0 (0.99–1.01)	0.984
Vacuum-assisted delivery	2.04 (0.69–6.03)	0.198
Medio-lateral episiotomy	0.61 (0.24–1.58)	0.313
Primiparity	0.78 (0.29–2.04)	0.611
Prolonged second stage	0.58 (0.19–1.77)	0.340
Occipito-posterior position	0.96 (0.2–4.62)	0.957

## Data Availability

Data supporting reported results can be found in a specific archived database generated during the study.

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
