# Peer review of "High Incidence of Obstetric Anal Sphincter Injuries among Immigrant Women of Asian Ethnicity"

_jcm, 2023, doi:10.3390/jcm12031044_

Round 1

Reviewer 1 Report

This manuscript compares the prevalence of OASI in Caucasian and women of Asian ethnicity in multiple centres in Germany. This is a useful and interesting study that supports previous findings around the high rate of OASI in Asian women. Overall it is generally well written. However, there are significant spelling errors throughout that detract from the good work presented within the manuscript.

The use of “Asian descent” a little cumbersome and suggest if it may be better to describe the women as per their ethnicity? ie women of Asian ethnicity.

It would be interesting for the reader to see if there was significance between 3rd and 4th degree OASI individually as well as combined.  

In the discussion the authors comment of the significance of language and communication. Is it possible to add who spoke local language to these data?

The analysis is very basic and with such good numbers would it be possible to do a multi regression analysis to identify the difference in the risk factors (if any) between Caucasian OASI group and Asian OASI group? The authors have introduced these factors into the discussion and it would not be outside the scope of the main paper.

Within the discussion take care with terminology “risk”. This suggests a risk analysis has been done rather than a difference in prevalence.  

Author Response

Reviewer 1:

  1. 1. The use of “Asian descent” a little cumbersome and suggest if it may be better to describe the women as per their ethnicity? ie women of Asian ethnicity.

Thank you for this comment. We made the requested change through the manuscript.

  1. 2. It would be interesting for the reader to see if there was significance between 3rd and 4th degree OASI individually as well as combined.

The requested data is presented in Table 1.

  1. 3. In the discussion the authors comment of the significance of language and communication. Is it possible to add who spoke local language to these data?

A very interesting question. Unfortunately, we don’t have the requested information, however communication with most Asian women was in English, which is usually not the native language of both patient and midwife. We clarified this issue in the Discussion section.

  1. The analysis is very basic and with such good numbers would it be possible to do a multi regression analysis to identify the difference in the risk factors (if any) between Caucasian OASI group and Asian OASI group?

A multivariable logistic regression analysis of various risk factors for 4th degree OASI is presented in Table 3. Of all obstetric parameters, only Asian ethnicity was found to be significantly associated with 4th degree OASI (OR 2.9; 95% C.I. 1.02 -8.35). 

  1. Within the discussion take care with terminology “risk”. This suggests a risk analysis has been done rather than a difference in prevalence.

The requested change was made.

Reviewer 2 Report

The author present interesting study regarding the obstetric anal sphincter injuries among Asian women living in a western countries. They underline underestimated problem of the OASI - higher rates of fourth-degree tears. Moreover authors found that Asian ethnicity was also associated with a significantly higher 22 risk for blood transfusion following OASI, and a lower tendency for postpartum follow-up.

The authors concisely summarize the current knowledge on possible OASI and clinical links between higher rate of OASI and higher rates of fourth-degree tears among Asian women. 

Minor errors:

1.    Text should be revised carefully once again, because some of sentences need to be revised due to used colloquialisms

In my opinion presented manuscript is well written, sum up an important clinical issue.

Author Response

Reviewer 2:

Thank you for your comments. The spelling errors have been corrected and several paragraphs were re-edited.